# Comparison between Ecological Momentary Assessment and Questionnaire for Assessing the Frequency of Waking-Time Non-Functional Oral Behaviours

**DOI:** 10.3390/jcm11195880

**Published:** 2022-10-05

**Authors:** Rosaria Bucci, Daniele Manfredini, Francesca Lenci, Vittorio Simeon, Alessandro Bracci, Ambrosina Michelotti

**Affiliations:** 1Department of Neurosciences, Reproductive Sciences and Oral Sciences, Section of Orthodontics and Temporomandibular Disorders, University of Naples “Federico II”, 80131 Naples, Italy; 2Department of Medical Biotechnologies, School of Dentistry, University of Siena, 53100 Siena, Italy; 3Department of Mental Health and Preventive Medicine, Medical Statistics Unit, University of Campania “Luigi Vanvitelli”, 81100 Naples, Italy; 4Department of Neuroscience, School of Dentistry, University of Padova, 35128 Padua, Italy

**Keywords:** ecological momentary assessment, Oral Behaviour Checklist, questionnaire, self-report, awake bruxism

## Abstract

Over the years, several tools have been proposed to measure oral behaviours (OB). Recently, a smartphone-based application for ecological momentary assessment (EMA) has been introduced to collect real-time data on waking-time OB. The aim of this study was to compare the self-reported frequency of OB by means of a standardised questionnaire with that recorded with a smartphone-based application for EMA. A total of 151 participants, recruited from the general population, were invited to fill in the Oral Behaviour Checklist (OBC). Scores for four questions concerning grinding, clenching, tooth contact, and mandible bracing were computed. Afterwards, participants were provided with a smartphone application for prolonged real-time reporting of OB. One-way analysis of variance (ANOVA) and a general linear mixed model (GLMM) were used to compare the responses to each OBC question with the frequencies of the same condition recorded with the EMA. Results showed significant association between OBC responses and the EMA recordings. In particular, increased frequencies of clenching, grinding, and teeth contact were recorded by individuals who provided higher OBC scores. On the other hand, a nonlinear association was observed for “mandible bracing”, pointing out difficulties in the comprehension of this condition.

## 1. Introduction

Oral parafunctional behaviours are activities of the stomatognathic system which go beyond normal oral functions. They are characterised by non-physiological jaw movements that often result in unnecessary overuse of the masticatory muscles [1,2]. Both functional and non-functional activities are listed among parafunctional behaviours; functional activities represent activities related to normal jaw functions which are performed to a higher extent, such as chewing, talking, and yawning, while non-functional oral activities include clenching or grinding the teeth, or holding the jaw rigid [3]. Long-term observational studies on large cohort of individuals have recognized that oral behaviours, including physiological and non-physiological functions of the masticatory system, represent the single most significant predictor of early-onset TMD, and play a major role in the persistence of painful TMD [4,5]. Furthermore, prolonged tooth-to-tooth contact and other parafunctional activities might be responsible for mechanical wear, loss of hard dental tissues, and increased interdental sensitivity [6,7,8].

Over the years, several tools have been proposed to record and measure oral behaviours (OB). Self-reports (questionnaires or checklists) completed by patients are tools often suggested to collect information on OB, especially concerning those activities performed during waking hours [6]. Among the available tools, the Oral Behaviour Checklist (OBC) has been considered a valid instrument quantifying the frequency of self-reported wake-time and night-time OB during the preceding month [8]. In particular, this tool has been developed to measure a wide range of activities including both functional (i.e., chewing, talking, yawning, singing) and non-functional (clenching, grinding, keeping teeth in contact, holding the mandible) activities. Thanks to its simple use, this written questionnaire, that has been included in the diagnostic criteria for TMD (DC/TMD), has been translated and adapted for use in different languages in order to apply it in research and clinical settings [9,10,11]. Studies have suggested that the assessment accuracy of self-reported OB might be low, especially concerning bruxism-related activities, due to a lack of individual awareness about such behaviours [12]. However, although relying on the attention/memory of the respondent, the construct validity of the OBC has been successfully verified for most of its items against surface electromyography (EMG), which is the only instrument providing an objective measurement of the masticatory muscle activity [13].

Ecological momentary assessment (EMA) was developed in 1980 as a strategy to overcome the limitations of traditional quantitative methods in psychological sciences [14,15]. This technique provides real-time patient reporting of the variable under investigation at multiple timepoints in the natural environment for prolonged periods of observation [16]. Based on that, smartphone app technology has been considered as an ideal platform for the adoption of EMA, thus fitting perfectly with the need of maximizing compliance and simplicity [17]. On the other hand, some drawbacks of the EMA approach have been reported, such as the risk of being annoying for participants, and the possibility of significant missing data due to poor compliance [18]. A dedicated smartphone application has been recently developed to apply the EMA principles into the field of waking-time OB clinical research [19]. In particular, this instrument has been introduced to measure jaw activities included in the spectrum of the awake bruxism, such as clenching and grinding the teeth, keeping teeth in contact, and holding the mandible rigid. This smartphone application is based on real-time evaluations at multiple daily recording points over multiple-day spans. Authors have hypothesised that, in patient populations, this application could also act as a biofeedback strategy by implementing patients’ education on OB, helping individuals to develop awareness and knowledge of their own OBs that might have negative consequences for their health [19].

A previous study by Kaplan and co-workers compared the assessment of several OBs with the OBC with the EMA using a portable, handheld computer for OB recordings [20]. Differently from the previous study, a smartphone-based app does not require dedicated experimental devices, but instead can be used on regular smartphones that are already part of the daily life of the vast majority of adult individuals. Therefore, the aim of this study was to compare the self-reported data of waking-time non-functional OBs by means of a retrospective single-point observation via the OBC, with the prolonged report of the same activities using the EMA approach with a dedicated smartphone-based application.

The null hypothesis was that EMA was not able to measure the same amount of bruxism-related OB as that reported with the OBC.

## 2. Materials and Methods

### 2.1. Participants’ Recruitment

The sample was recruited from the general population of the city of Naples (Italy), by means of direct contact via email or via WhatsApp, between March 2020 and June 2020. Adult individuals (≥18 years of age), willing to participate in the study, having a smartphone, and presenting with good general health, were enrolled. Incomplete records were excluded. This study was conducted in accordance with the Helsinki Declaration, and was approved by the local ethical committee of the University of Naples Federico II (Italy, protocol approval n. 420/20). Volunteers willing to participate were provided with written standardised instructions and with an ID code for anonymization of the data.

### 2.2. Assessment of Waking-State Oral Behaviours

#### 2.2.1. Questionnaire

The full-version OBC was provided (21 items). Each item aimed to assess the frequency of one specific OB, in the preceding month (last 30 days). Response options were based on a 5-point Likert scale as follows: “none of the time” (0), “a little of the time” (1), “some of the time” (2), “most of the time” (3), “all of the time” (4) [19]. For the purpose of the current study, only 4 questions were considered to be of interest:-Q3: Grind teeth together during waking hours.-Q4: Clench teeth together during waking hours.-Q5: Press, touch, or hold teeth together other than while eating (that is, contact between upper and lower teeth).-Q6: Hold, tighten, or tense muscles without clenching or bringing teeth together.

#### 2.2.2. Ecological Momentary Assessment (EMA)

An application for smartphone, “BruxApp^®^” (WMA srl, Italy), was used [17]. This application has been designed to send 20 alert sounds at random hours during the day, in order to collect data on self-reported jaw activities. Participants were taught to focus on their current condition and to answer the alert within 5 min from the alert sound by tapping on the display icon that refers to the current condition of the jaw. The following conditions were listed: relaxed jaw muscles; teeth contact; teeth clenching; teeth grinding; mandible bracing (without teeth contact). After 5 min, answers could not be stored in the software, and an error message appeared on the display. Participants were asked to ignore the alert if it appeared while eating, talking, or driving. Recording time was set automatically from 8.00 to 12.30 and from 14.30 to 22.00 every day, to reduce the possibility that alert sounds were generated during mealtimes. A minimum of 12 alert answers per day was required to record the day as “valid”. In case of failure to reach the minimum threshold to validate the day, the software automatically set an additional recording day to complete a 7-day protocol. After the 7 valid days of recording, the software generated a pre-formatted spreadsheet including the responses for each day (both “valid” and “not valid”). The participants were invited to forward the report spreadsheet via email.

#### 2.2.3. Satisfaction with the Tool

Once the study period was successfully completed, all participants received a satisfaction questionnaire composed by six items concerning their experience with both BruxApp^®^ and OBC tools. The satisfaction questionnaire addressed three domains: time (Q1: “Filling in the OBC questionnaire took a large amount of time”; Q2: “Using the BruxApp^®^ took a large amount of time”), interference with daily activities (Q3: “Filling in the OBC questionnaire interfered with my daily activities”; Q4: “Using the BruxApp^®^ interfered with my daily activities”), and awareness (Q5: “Filling in the OBC questionnaire I became more aware of oral behaviours I was not aware of”; Q6: “Using the BruxApp^®^ I became aware of oral behaviours I was not aware of”). Each question could be answered on a 10-point scale ranging from 0 (totally disagree) to 10 (totally agree).

#### 2.2.4. Statistical Analysis

Descriptive statistics on age, gender, OBC responses, and EMA responses were computed. Continuous variables were reported as means and standard deviations, while categorical variables were reported as absolute numbers and percentages. One-way analysis of variance (ANOVA) was used to compare the mean percentage of responses per condition according to the EMA, with the responses of the individual items of the OBC. In particular, Q3 was computed with the response option “Grinding”; Q4 was computed with the response option “Teeth Clenching”; Q5 was computed with the response option “Teeth Contact”; and Q6 was computed with the response option “Mandible Bracing (no teeth contact)”. Furthermore, ANOVA was also used to measure differences in the reported frequency of each OB with EMA during the 7-day period. A general linear mixed model (GLMM) was used to assess the association of the reported frequency of each OB measured with the EMA on different days with OBC-specific test scores. Contrasts of marginal linear prediction with chi2, df e *p* value were reported. Finally, a paired-sample t-test was performed to compare the scores of each domain of the satisfaction questionnaire between the two tools (OBC vs. EMA). The level of statistical significance was set at *p* < 0.05. Statistical analysis was performed using STATA version 14.0 (StataCorp LP).

The sample size was calculated by targeting the accuracy of the response “none of the time” for the questions in the OBC analysis. The initial precision (95% confidence interval) was very high with a width of 0.12, produced by a sample size of 281 and a conservative sample proportion of 0.5. After the removal of incomplete data, it was noted that the confidence level width was around 0.15 and considered it acceptable for the evaluation.

## 3. Results

### 3.1. Sample

A total of 276 individuals were contacted to participate in the study. After excluding those who did not answer and those with incomplete records, 151 participants were included in the final sample (99 females, 52 males; mean age 27.2 ± 8.1 years) (Figure 1).

### 3.2. Assessment of Waking-State Oral Behaviours

The frequencies of self-reported OB as referred with the OBC are reported in Table 1. The most frequent response option for Q3 and Q6 was “none of the time” (70.2% and 47.0%, respectively), while the most frequent response option for Q4 and Q5 was “little of the time” (43.0% and 44.3%, respectively).

Assuming the threshold of 12 answered alerts per day to record a valid day, the mean number of days necessary to complete the 7-day protocol with EMA was 9.3 ± 2.4 (range 7–22) (Table 2). The most frequent number of days needed was 8 (31.7%). Approximately 65% of the participants completed the 7-day protocol within 9 days, while the remaining 35% completed the recording in more than 9 days.

The mean percentages of the frequency of different OBs recorded with EMA are shown in Table 3. The most frequent condition was “relaxed” (62.5%), followed by “teeth contact” (18.8%).

During the 7-day period of recording, the ANOVA pointed out nonsignificant changes in the amount of individual activities recorded (relaxed—F(6912) = 1.42, *p* = 0.204; mandible bracing—F(6912) = 0.62, *p* = 0.718; teeth contact—F(6912) = 1.31, *p* = 0.249; grinding—F(6912) = 1.88, *p* = 0.082; clenching—F(6912) = 0.51; *p* = 0.082) (Table 4).

### 3.3. Relationship between OBC and EMA Responses

For all the study variables, a significant association was observed between the self-assessed frequency of OBs reported with the OBC and the respective condition recorded with EMA over the 7-day period (Q3 vs. grinding—F(3147) = 56.21, *p* = 0.000; Q4 vs. clenching—F(3147) = 11.84, *p*= 0.000; Q5 vs. teeth contact—F(4146) = 5.60, *p* = 0.000; Q6 vs. mandible bracing—F(4146) = 4.13, *p* = 0.003). In particular, a linear increase in responses was observed for Q3, Q4, and Q5, meaning that the higher the score provided with the OBC, the higher the frequencies of teeth grinding, teeth clenching, and teeth contact that was recorded with the EMA (Figure 2). On the contrary, inconsistent reports were observed for the condition “mandible bracing” compared with Q6 (Figure 2).

Regarding the “grinding” activity recorded with the EMA on different days, the GLMM showed a significant effect of day, OBC score (Q3), and of the interaction Day*Q3. Concerning the “clenching” activity, a significant effect of day and OBC score (Q4) was found. For both “teeth contact” and “mandible bracing”, a significant effect of OBC score only was observed (Table 5).

Finally, a comparison was made between the sum of the four OBC questions (OBC sum) and the overall conditions recorded with EMA during the 7-day period excluding the “relaxing” condition. The OBC sum was divided into 3 categories: 0 = if the OBC sum was equal to 0; 1 = if OBC sum was higher than or equal to 1 and lower than or equal to 4; 2 = if OBC sum was higher than or equal to 5. Statistically significant differences in the percentage of EMA responses according to the OBC sum categories was observed [F (2148) = 15.73; *p* = 0.0000] (Table 6).

### 3.4. Satisfaction with the Tool

A statistically significant difference was observed for all the three domains between the two tools. In particular, significantly less “Time” and “Interference” responses were reported with OBC as compared with EMA, while significantly increased “Awareness” was observed with EMA as compared with OBC (Table 7).

## 4. Discussion

The present cross-sectional study based on data gathered from the general population aimed to compare two different tools for the self-assessment of the frequency of oral behaviours (OBs): the Oral Behaviour Checklist (OBC) and the ecological momentary assessment (EMA) via a dedicated smartphone application (Bruxapp^®^). Overall, in the current sample, participants consistently reported similar frequency of OB with both instruments considering the activities of “Clenching”, “Grinding”, and “Teeth Contact”, but discrepancies were observed with regard to “Mandible Bracing”.

Both tools assessed in the current study were designed to collect self-reported information regarding waking-time OBs; OBC provided a retrospective single-point observation, while real-time data can be collected within a certain period of time in the natural setting with EMA. With the OBC, participants were asked to respond to the question “How often do you do each of the following activities, based on the last month?”; therefore, due to its retrospective nature, the questionnaire relies on patient’s memory. Furthermore, timing of the assessment may also influence the responses, as there is a fluctuation in these activities depending on the moment of day in which the questionnaire is completed [20]. On the other hand, EMA allows multiple recording of a given condition, but still relies on the attention and conscious awareness of the respondent [21].

With regard to the prevalence of OB, this survey showed that the most frequent condition detected with EMA was “Relaxed” (62.5%), while the less frequently detected was “Grinding” (0.5%). It has to be underlined that the present study was conducted during the COVID-19 pandemic period, which might have increased psychological distress, parafunctional activities, and TMD in the general population, according to some authors [22,23]. Notwithstanding, the prevalence observed in the current study sample was in accordance with the findings observed in similar previous studies conducted with EMA on young adults (university students) [24,25,26]. Similarly, the frequencies of OBs reported with OBC reflected the prevalence already reported in the general population [2].

Almost 30% of the recruited participants dropped out from the study as they could not download the smartphone application, or they did not complete the 7-day recording period. Furthermore, the respondents reported that EMA was significantly more time-consuming and significantly interfered more with their daily activities, as compared with the questionnaire. This is in contrast with the recent findings supporting the hypothesis that the smartphone-based EMA approach was easy to integrate into a daily routine [27]. This discrepancy could be explained by the differences in the study samples, since the participants of the previous study were all university students, while in the current survey the participants were recruited from the general population, including workers and employees, that most likely encountered the major difficulties in being compliant with the alerts during working hours. Furthermore, the participants of the current sample were not recruited from a patient population, meaning that they did not demand any treatment, and did not receive any kind of reward, thus reducing their motivation to successfully complete the experimental period. On the other hand, interestingly, participants reported that one-week EMA recording significantly increased their ability to recognise some OBs that they were not aware of, thus supporting the potential biofeedback role of the EMA. This finding is in contrast with the result of the statistical analysis assessing the response rate during the 7-day observation period, that pointed out no significant difference in the frequency of OBs during 7 days. Therefore, it seems that, although participants became more aware of their OBs, they did not change their habits during the observational period. This finding has two different interpretations: firstly, since the participants were recruited from the general population, they were not interested in taking action with a behavioural change; secondly, the main purpose of the smartphone application used in the current study was to measure the frequency of OBs, and no suggestions were given to the individuals about the potential harmfulness of some OBs and about the possibility of changing their oral activities. Therefore, a dedicated smartphone application for patients’ instruction about habit-reversal behavioural therapy should be implemented to provide a clinical contribution in the management of some temporomandibular disorders.

Considering the overall responses to the questionnaire, for future research, the OBC seems to be more indicated for screening purposes as it requires a short amount of time and does not interfere with one’s daily activities, while the EMA could be suggested for those patients who require a more extensive and in-depth assessment of bruxism-related OBs.

Comparing the frequency of each OB reported with the OBC with the mean % of the corresponding condition recorded with EMA over the 7-day period, the ANOVA pointed out statistically significant differences for all the studied variables. In particular, in the comparisons “Q3 vs. Grinding”, “Q4 vs. Clenching”, and “Q5 vs. Teeth contact”, a linear correlation between the two instruments was observed, meaning that individuals who reported high frequency of a given OB with the OBC also reported a greater percentage of response frequency of the same OB with the EMA. On the other hand, in the comparison “Q6 vs. Mandible Bracing”, the correlation between the response scales was not linear, thus supporting that “Mandible Bracing” might be an activity which is difficult to understand, and further clarification is needed. Two interesting findings have emerged from the comparison between questionnaire and EMA: Firstly, those individuals who were aware of their non-functional behaviours and reported high values on the OBC Likert-scale tended to overestimate the frequency of their non-functional behaviours, compared with the EMA recording. For instance, they reported to perform a given activity “most of the time” or “all of the time”, but instead the activity was recorded in less than 50% of the total EMA alerts. This could be explained by the fact that the “time” is interpreted in a different way by the two different tools. By filling in the questionnaire, participants are invited to provide the frequency of all OBs at the same time, while for each EMA alert sound, the respondent can select only one single condition. On the other hand, it was also observed that many individuals who were not aware of their non-functional activities and provided low scores in the OBC reported to perform a given OB in the 20% of the total EMA alerts (especially with regard to “Teeth Contact”).

OB assessment via OBC and EMA present two major differences. Firstly, the EMA allows a recording of the “relaxed” condition, which is not included in the OBC. Secondly, with the OBC, individuals are invited to report the frequency of all the conditions in a single-point assessment, while with EMA, only one condition at the time can be selected. For this reason, the sum of the OBC responses was compared with the overall conditions recorded with EMA, excluding the “Relaxed” condition. Additionally, this analysis showed a good overlapping of responses, meaning that those who reported overall lower degree of OB with the OBC also showed lower frequencies of non-relaxed conditions with EMA.

This study presented some limitations. Firstly, only one subject in the entire sample reported a high frequency of “Grinding” activity in the questionnaire, thus limiting the external validity of the statistics provided on this outcome. However, this reflects the low prevalence of this activity, which is more frequently observed during sleep rather than during waking hours [6]. Secondly, no clinical examination was performed; therefore, no information was collected on TMD signs and symptoms. It would be interesting in future studies to verify the correspondence of these two tools in a sample of patients.

## 5. Conclusions

In the current sample of adults recruited from the general population, the use of a standardised questionnaire (i.e., Oral Behaviour Checklist) correctly measured the frequency of non-functional oral behaviours during waking hours as compared with a real-time assessment through a smartphone-based application for ecological momentary assessment over a 7-day period.Due to the limited interference with the daily activities, Oral Behaviour Checklist can be suggested for screening purpose during initial consultations.Given the increased awareness of awake bruxism activities observed in the general population following the use of the smartphone-based app, studies are needed to test the role of this tool in behavioural therapy for patients with potential clinical consequences.

## Figures and Tables

**Figure 1 jcm-11-05880-f001:**
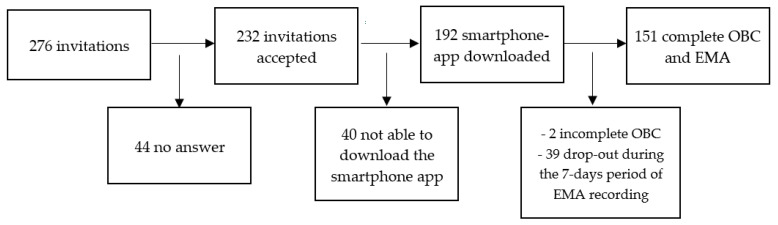
Flowchart of the participant-recruitment process. OBC: Oral Behaviours Checklist; EMA: Ecological Momentary Assessment.

**Figure 2 jcm-11-05880-f002:**
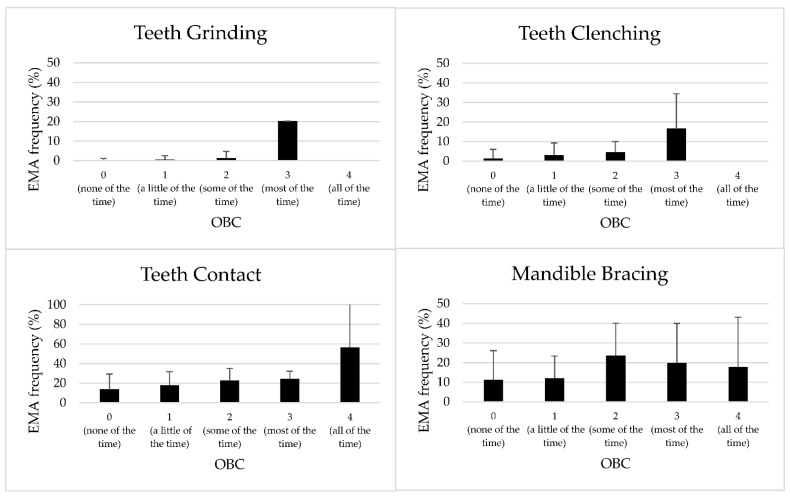
Mean frequency and standard deviation of each condition recorded with the smartphone-based application for Ecological Momentary Assessment (EMA) over a 7-day period, according to the different responses of the corresponding Oral Behaviour Checklist (OBC) questions.

**Table 1 jcm-11-05880-t001:** Frequency of responses for each Oral Behaviours Checklist (OBC) question.

	0 (None of the Time) N (%)	1 (Little of the Time) N (%)	2 (Some of the Time) N (%)	3 (Most of the Time) N (%)	4 (All of the Time) N (%)
Q3	106 (70.2%)	28 (18.5%)	16 (10.6%)	1 (0.7%)	0 (0)
Q4	44 (29.1%)	65 (43.0%)	34 (22.5%)	8 (5.3%)	0 (0)
Q5	38 (25.1%)	67 (44.3%)	35 (23.1%)	9 (5.9%)	2 (1.3%)
Q6	71 (47.0%)	41 (27%)	27 (17.8%)	10 (6.6%)	2 (1.3%)

**Table 2 jcm-11-05880-t002:** Frequency of days required to record a full valid 7-day observation period.

Days	Frequency %
7	17.8%
8	31.7%
9	15.2%
10	12.5%
11	6.6%
12	7.2%
13	4.6%
15	0.6%
16	1.3%
19	0.6%
20	0.6%
21	0.6%
22	0.6%

**Table 3 jcm-11-05880-t003:** Frequency of each condition recorded with the Ecological Momentary Assessment (EMA) over the 7-day observation period. Data are shown as mean percentages ± standard deviation (SD).

	Mean (%)	SD
Relaxed	62.5%	26
Teeth Clenching	3.6%	7.3
Teeth Contact	18.8%	15.2
Teeth Grinding	0.5%	2.2
Mandible Bracing	14.3%	15.3

**Table 4 jcm-11-05880-t004:** Mean percentage of frequencies and confidence intervals (C.I.) of Oral Behaviours (OB) recorded with the Ecological Momentary Assessment (EMA) at different timepoints.

EMA Recording Day	Relaxed		Mandible Bracing		Teeth Contact		Grinding		Clenching	
	Mean	C.I.	Mean	C.I.	Mean	C.I.	Mean	C.I.	Mean	C.I.
1	61.36%	59.14–63.58	14.59%	13.03–16.14	20.57%	18.78–22.36	2.88%	1.89–3.86	0.58%	0.26–0.91
2	60.67%	58.45–62.89	14.74%	13.18–16.29	19.46%	17.67–21.24	4.7%	3.72–5.69	0.36%	0.04–0.69
3	61.93%	59.72–64.15	14.62%	13.07–16.17	18.86%	17.07–20.65	3.83%	2.84–4.81	0.63%	0.3–0.95
4	63.96%	61.74–66.17	13.9%	12.34–15.45	18.64%	16.85–20.42	3.12%	2.13–4.10	0.48%	0.15–0.81
5	63.1%	60.88–65.32	13.36%	11.81–14.91	18.63%	16.85–20.42	4.24%	3.25–5.22	0.64%	0.31–0.97
6	63.69%	61.47–65.9	14.99%	13.44–16.54	17.09%	15.31–18.88	3.79%	2.8–4.77	0.42%	0.97–0.75
7	63.98%	61.76–66.2	13.65%	12.1–15.2	18.7%	16.91–20.48	2.98%	1.99–3.96	0.67%	0.34–0.99

**Table 5 jcm-11-05880-t005:** Results of the General Linear Mixed Model (GLMM) for repeated measurements assessing the association of the reported frequency of each Oral Behaviour (OB) activity recorded with the smartphone-based application for Ecological Momentary Assessment (EMA) on different days, with the specific Oral Behaviours Checklist (OBC) test scores. Statistically significant differences are reported in bold.

		df	Chi2	*p*-Value
Teeth Griding	Day	6	214.80	**<0.001**
	Q3	3	176.33	**<0.001**
	Day × Q3	18	262.23	**<0.001**
Teeth Clenching	Day	6	20.56	**0.002**
	Q4	3	35.77	**<0.001**
	Day × Q4	18	26.11	0.097
Teeth Contact	Day	6	9.12	0.166
	Q5	4	22.00	**<0.001**
	Day × Q5	24	22.25	0.564
Mandible Bracing	Day	6	6.55	0.364
	Q5	4	16.25	**0.003**
	Day × Q5	24	27.04	0.302

**Table 6 jcm-11-05880-t006:** Mean percentages and standard deviation (SD) of frequencies of “Non-Relaxed” condition recorded with the Ecological Momentary Assessment (EMA), according to the Oral Behaviours Checklist (OBC) sum categories.

OBC Sum Category	Mean %	SD
0	25.24%	25.7
1–4	31.03%	22.7
>4	53.13%	25.1

**Table 7 jcm-11-05880-t007:** Scores of the three domains of the satisfaction questionnaire, concerning the two tools adopted for the Oral behaviours (OB) measurement, and results of the paired-sample t-test. Data are shown as mean ± standard deviation. OBC: Oral Behaviours Checklist; EMA: Ecological Momentary Assessment.

	OBC	EMA	*p*-Value
Time	3.15 ± 2.2	3.70 ± 2.3	0.017
Interference	2.39 ± 1.8	3.37 ± 2.3	0.000
Awareness	4.92 ± 2.9	5.8 ± 2.9	0.001

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
