# Peer review of "Comparison between Ecological Momentary Assessment and Questionnaire for Assessing the Frequency of Waking-Time Non-Functional Oral Behaviours"

_jcm, 2022, doi:10.3390/jcm11195880_

Round 1

Reviewer 1 Report (New Reviewer)

Thank you very much for the opportunity to review the manuscript: “Comparison between Ecological Momentary Assessment and Questionnaire for the Assessment of Frequency of Waking-Time Non-Functional Oral Behaviours”. The subject of this article is very interesting, and up-to-date. However, I have some questions and would like to point some details that should be corrected:

1.     Version submitted to the revision contain some parts mark red. Is it a resubmission of previously rejected article or it has already been reviewed and some changes are introduced?

2.     The article would benefit from a correction by a native speaker. 

3.     The references should be corrected. In the introduction there are multiplied references [1,2] (after “overuse the masticatory muscles”). The list of references is not prepared according to the recommendations for authors. 

4.     Tables and figures should be placed in the text near the information provided, not at the end of the article.

5.     Please provide the developer of the BruxApp application

Author Response

Reviewer 1

Thank you very much for the opportunity to review the manuscript: “Comparison between Ecological Momentary Assessment and Questionnaire for the Assessment of Frequency of Waking-Time Non-Functional Oral Behaviours”. The subject of this article is very interesting, and up-to-date. However, I have some questions and would like to point some details that should be corrected:

  1. Version submitted to the revision contain some parts mark red. Is it a resubmission of previously rejected article or it has already been reviewed and some changes are introduced?
    • We thank the author for the question. This is a resubmission of a previously review (not rejected) article. The text marked in red highlighted the changes according to the previous comments or suggestions
  2. The article would benefit from a correction by a native speaker. 
    • The article has been further reviewed for style and grammar. Thank you!
  3. The references should be corrected. In the introduction there are multiplied references [1,2] (after “overuse the masticatory muscles”). The list of references is not prepared according to the recommendations for authors. 
    • Thank you for the accurate comment. The multiple references [1,2] have been deleted. Regarding the list of references, the citation style has been updated using the template tool provided by MDPI.
  4. Tables and figures should be placed in the text near the information provided, not at the end of the article.
    • Tables and figures have been correctly moved. Thank you.
  5. Please provide the developer of the BruxApp application
    • The developer of the Bruxapp application has been added.

Reviewer 2 Report (New Reviewer)

1. Introduction: Add the null hypothesis of the study.

2. Materials and Methods: Was there any calibration for the analysis?

3. Materials and Methods: Add the manufacturer name of the STATA version 14.0 program.

4. Results: How was the sample size calculated?

5. Discussion Describe clinical contribution of the study and directions for future research in the field.

Author Response

Reviewer 2:

  1. Introduction: Add the null hypothesis of the study.
    • The null hypothesis was added in the introduction. Thank you for the suggestion.
  2. Materials and Methods: Was there any calibration for the analysis?
    • Thank you for your question. For the current study, no calibration was performed as both instruments (OBC and Bruxapp) have been previously tested for their validity and widely used in numerous studies to measure what they are meant to measure (oral behaviours).
  3. Materials and Methods: Add the manufacturer name of the STATA version 14.0 program.
    • Amended, thank you.
  4. Results: How was the sample size calculated?
    • The sample size calculation has been added to the methods as follows “The sample size was calculated by targeting the accuracy of the response 'none of the time' for the questions in the OBC analysis. The initial precision (95% confidence interval) was very high with a width of 0.12, produced by a sample size of 281 and a conservative sample proportion of 0.5. After the loss of the participants, we noted that the confidence level width was around 0.15 and considered it acceptable for the evaluation.”

  1. Discussion Describe clinical contribution of the study and directions for future research in the field.
  • Thank you for the suggestion. Clinical and research inputs have been added to the discussion.

This manuscript is a resubmission of an earlier submission. The following is a list of the peer review reports and author responses from that submission.

Round 1

Reviewer 1 Report

This article compares waking-time non-functional oral behaviors assessed through the OBC with a smartphone-based application for EMA. The authors hypothesize states that the smartphone-based application could also act as a biofeedback strategy by implementing patients’ education on OB, and helping individuals to develop awareness and knowledge of their own OB that may be negative for their health. The authors conclude that the OBC correctly measured the frequency of non-functional oral behaviors during waking hours as compared to real-time assessment through smartphone-based application for Ecological Momentary Assessment over a 7-days period. In addition, the authors claim that Ecological Momentary Assessment represents a promising approach in the behavioral therapy to implement patients’ awareness of oral behaviors. Overall, this manuscript is lacking details in its’ rationale and methodology that should be considered.

1) Given that the study was conducted among the general population and not patients or individuals under therapy for OB or TMD, the author’s conclusions claim that that Ecological Momentary Assessment represents a promising approach in the behavioral therapy to implement patients’ awareness of oral behaviors appears to be a stretch. Moreso, not only was the sample a non-patient sample, methods of recruitment via direct contact through email/WhatsApp suggest a potential biased sample, if not a convenient sample.

2) It is unclear what exactly is the rationale of this study and why was this study conducted? Given that the Kaplan et al paper determines the validity of the OBC using EMA, how does this study build on the use of EMA through a smart-phone application for assessment and further investigation of behaviors?

3) It is not clear to this reviewer as to why questions on “Time” and “Interference” were assessed? Given the increased “Interference” with EMA, how likely is it that the reported increased “Awareness” is confounded by the increased “Interference”?

4) The authors use a one-way ANOVA to measure differences in the reported frequency of each OB with EMA during the 7-days period. Given the design that participants provided repeated measures, is using a one-way ANOVA the correct statistical test? Authors are advised to consult with a statistician.

5    5) Lastly, there a several places in the manuscript where punctuation is missing and sentences have grammatical errors.

Reviewer 2 Report

1.How did the authors ensure patient compliance especially in the App group?

2.Further comments are mentioned in the attached manuscript.
